# Ergonomic assessment of work-related musculoskeletal disorder and its determinants among commercial mini bus drivers and driver assistants (mini bus conductors) in Nigeria

Echezona Nelson Dominic Ekechukwu[1,2,3]*, Erobogha Useh[1], Obumneme Linky Nna[1], Nmachukwu Ifeoma Ekechukwu[2], Ogbonna Nnajiobi Obi[4], Emmanuel Nwabueze Aguwa[3,5], Sussan Uzoamaka Arinze-Onyia[6], Ukachukwu Okaroafor Abaraogu[1,7], Victor Adimabua Utti[8]

1 Department of Medical Rehabilitation, Faculty of Health Sciences and Technology, College of Medicine, University of Nigeria, Enugu, Nigeria, 2 LANCET Physiotherapy, Wellness and Research Centre, Enugu, Nigeria, 3 Environmental and Occupational Health Unit, Institute of Public Health, College of Medicine, University of Nigeria, Enugu, Nigeria, 4 Department of Physiotherapy, University of Port Harcourt Teaching Hospital, Port Harcourt, Nigeria, 5 Department of Community Medicine, Faculty of Medical Sciences, College of Medicine, University of Nigeria, Enugu, Nigeria, 6 Department of Community Medicine, College of Medicine, Enugu State University, Parklane, Enugu, Nigeria, 7 Physiotherapy Department, Glasgow Caledonian University, Glasgow, United Kingdom, 8 School of Sport, Rehabilitation and Exercise Sciences, University of Essex, Colchester, United Kingdom

* nelson.ekechukwu@unn.edu.ng

## Abstract

### Introduction

Work-related musculoskeletal disorder (WMSD) is a leading causes of occupational injury and disability among drivers and workers in the transport industry. This study evaluated the ergonomically assessed WMSD and its determinants among Nigerian commercial mini bus drivers (BD) and mini bus conductors (BC)

### Method

A total of 379 participants (BD = 200, BC = 179) were purposively sampled for this exploratory cross-sectional study. Participants' WMSD and work related variables were respectively assessed using the standardized Nordic questionnaire and a content-validated, Driving Work Station Assessment (DWSA) form. Data were analyzed descriptively and inferentially using chi-square and logistic regression. The level of significance was set at α = 0.05.

### Results

The participants were aged between 20 and 66 years, with a mean age of 33.26 ±10.76years (BD = 38.42±10.22years, BC = 27.50±8.13years); most of whom consumed alcohol (84.4%) and experienced severe job stress (73.4%). There was a high prevalence

**Data Availability Statement:** The data underlying the results presented in the study are available

from (Mendeley Data Repository - 10.17632/
brt3myjxbm.1). EKECHUKWU, Echezona Nelson
Dominic (2021), "WMSD among Bus Drivers and
Conductors", Mendeley Data, V1, doi: 10.17632/
brt3myjxbm.1

**Funding:** The author(s) received no specific
funding for this work.

**Competing interests:** The authors have declared
that no competing interests exist.

(95.8%; BD = 94.5%, BC = 97.8%) of WMSDs, the lower back (66.8%) and upper back (54.1%) had the highest regional prevalence of WMSD. The BC (BC vs BD) had significantly (p<0.05) higher prevalence of Neck (47.7% vs 21.5%) and upper back (80.4% vs 30.5%) WMSDs. Conversely, the BD (BD vs BC) had significantly (p<0.05) higher prevalence of low-back (85.0% vs 46.4%), knee (25.0% vs 9.5%), elbow (11.5% vs 3.9%), and wrist (10.5% vs 3.4%) WMSD. There was a significant association between WMSD and each of work duration ($X^2$ = 11.634, p = 0.009), work frequency ($X^2$ = 8.394, p = 0.039), job dissatisfaction (X2 = 10.620, p = 0.001) and job stress ($X^2$ = 16.879, p = 0.001). Working beyond 4days/week (OR = 10.019, p = 0.001), job dissatisfaction (OR = 1.990, p = 0.031), constrained working postures (OR = 5.324, p = 0.003) and fatigue (OR = 4.719, p = 0.002) were the predictors of WMSD.

## Conclusion

Job stress, work duration and work frequency, posture and fatigue are important determinants of WMSDs among mini bus drivers and their assistants in Nigeria. Ergonomics training intervention for this population is recommended.

## Introduction

Work related Musculoskeletal disorders (WMSDs) include a wide range of inflammatory and degenerative conditions affecting the muscles, tendons, ligaments, joints, peripheral nerves, and supporting blood vessels, that impact on the quality of work life [1]. These disorders are usually progressive and are associated with pains. It is known that WMSDs can affect several parts of the body including upper and lower back, spine, neck, shoulders and extremities [2, 3]. It has been opined that WMSD is related to the kind of occupation one is involved in, such as driving, manual handling, as well as awkward body postures and gestures [4, 5].

Commercial mini-bus driving task in Nigeria is associated with prolonged sitting or standing. This practice may increase the risk of developing WMSDs, and this may be further heightened when such task is performed in an ergonomically mismatched environment such as poor seat dimensions, abnormal reach envelope, excessive vibration etc (Fig 1). During prolonged sitting, there is an increase in the pressure on the ischial tuberosities from the upper body weight [6]. The resolution of these force results in an increase in spinal loading especially on the lumbar ligaments and intervertebral discs capable of causing some micro-trauma. The cumulative trauma caused from a sustained static loading on the soft tissues of the lumbar spine results in tissue damage and release of metabolites such as prostaglandins, histamine, serotonins and other cytokines that further irritate the surrounding soft tissue that may results in paraspinal muscle spasm and hyperexcitability [7]. A chronic static loading may also accelerate disc degeneration and herniation. It is the summation of all these biomechanical and biochemical processes that finally result in WMSD from prolonged sitting. On the other hand, prolonged standing can cause fatigue, leg cramps and backache due to venous insufficiency [8]. This may be attributed to the squeezing effect of the leg muscles on the deep veins like a tourniquet that results in reduced venous return, reduced cardiac output, reduced tissue perfusion and finally, increased tendency of tissue damage (WMSD).

More worrisome are the commonly observed work gestures and postures of the commercial mini-driver-assistants (popularly known as bus conductors), in Nigeria. In a bid to maximize

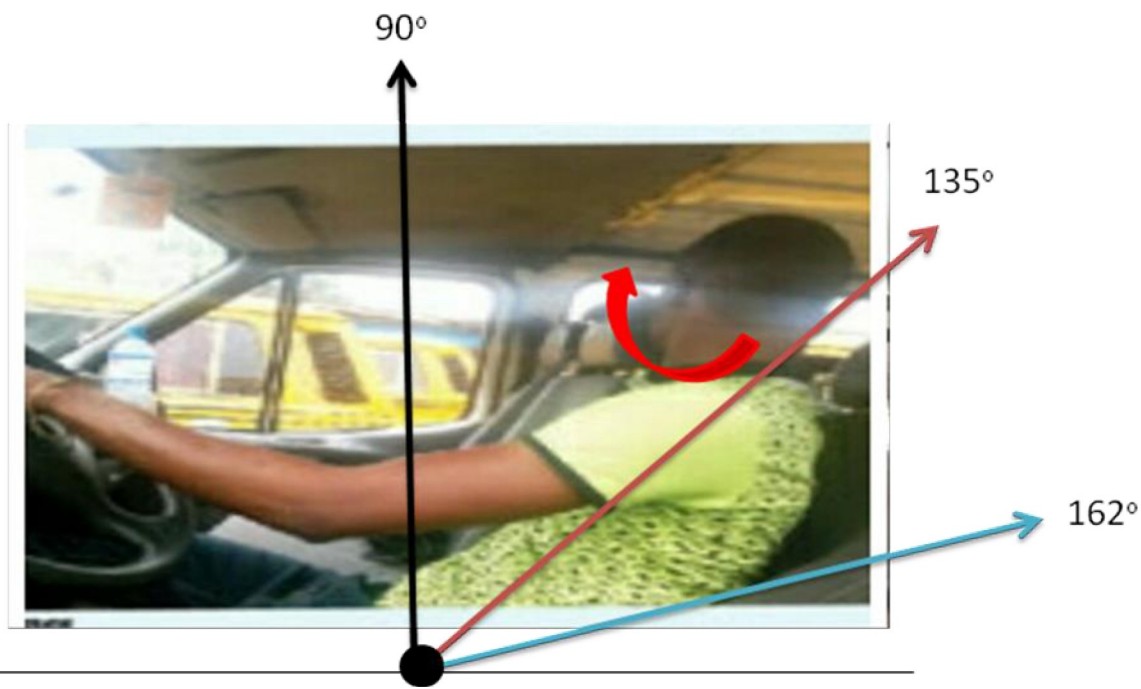

**Fig 1. A commercial bus driver using a poorly adjusted / damaged seat.**

profits possibly due to the prevailing economic crunch, many of these mini bus-conductors often give up their seats to passengers while they "hang"—a slogan used to describe the act of standing on the mini-bus' entrance while the bus is in motion (Fig 2). It has been recently

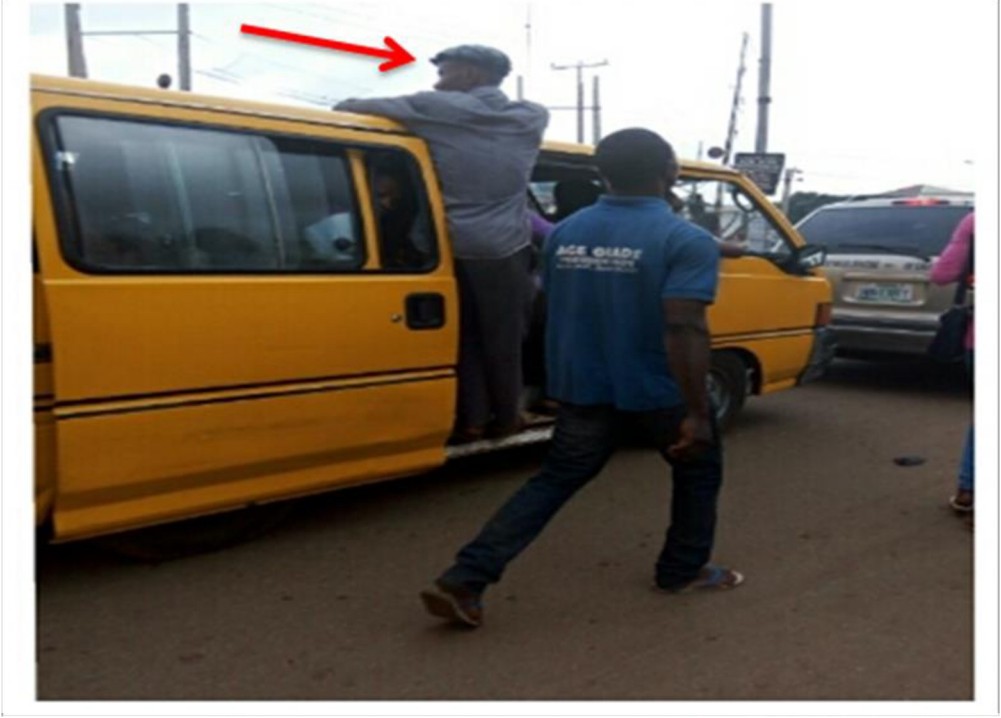

**Fig 2. A bus conductor standing at the bus' entrance while on motion.**

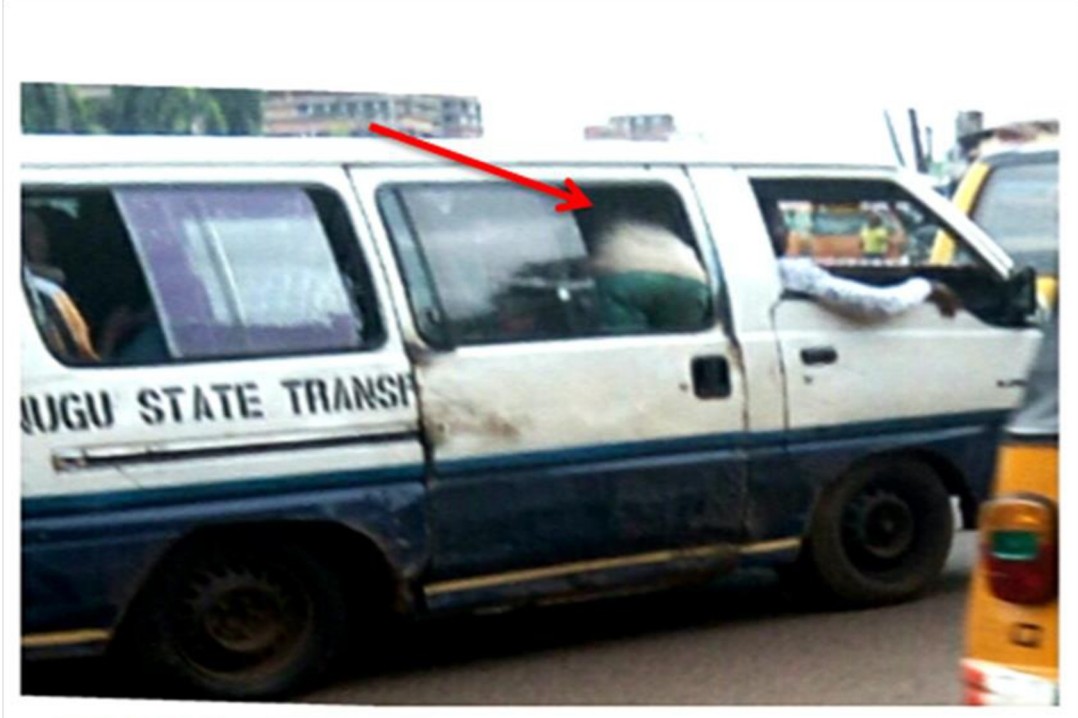

**Fig 3. A typical "Trunk-Twist" posture of bus conductors.**

observed that, due to the increased risk of accident associated with "hanging" and arrest by traffic law enforcement agents, the mini-bus conductors stand inside the mini-bus bent over (a full bow posture) with the slide door closed behind them while they perform their task (Fig 3). This prolonged awkward posture is feared to culminate in WMSDs among these cohorts. Working with a bent and/or twisted trunk can result in an overload of the spinal structures and increased activity of the entire muscles [9], thereby resulting in spinal disorders, as well as musculoskeletal pain.

Several studies have reported the prevalence of WMSDs among bus drivers. In a study by Szeto, and Lam [10], on work-related musculoskeletal disorders in urban drivers in Hong Kong, they reported higher MSDs around the neck, back, shoulder and knee/thigh regions that ranged between 30% and 60%. They also reported prolonged sitting and anthropometric mismatch as the most related causes of the musculoskeletal discomfort. Also in the study by Yasobant et al [11], that assessed the risks of developing WMSDs among bus drivers, they reported major musculoskeletal problems in the neck, back, upper limbs, knees and ankles, they also reported unsafe ergonomic practices and health risks as the primary causes of WMSD. In a similar study in Nigeria by Onawumi et al [12], and Akinpelu et al [2], on the prevalence of WMSD among occupational taxicab drivers, they found a high WMSD prevalence among their participants. Literature on transport ergonomics and WMSD prevalence in this industry are sparse and largely focused on bus drivers; thus, a dearth of studies among the bus conductors. Secondly, there appear to be no study that has compared WMSDs between bus drivers and bus conductors globally. Finally, there are few, yet inconsistent reports on the WMSD determinants among bus operators. It was hypothesised that (i) the WMSD prevalence among commercial mini-bus conductors will be significantly higher than those of the mini-

bus drivers, (ii) posture, stress and fatigue will be significant determinants of WMSD among commercial mini-bus drivers and conductors.

## Method

### Participants

Only commercial mini-bus operators (drivers and conductors) aged 18 years and above, who operated within Enugu Metropolis and had been on the job for at least one year, were included in this study. However, subjects with any obvious cofounding musculoskeletal deformities such as scoliosis, kyphosis etc were excluded from the study. The minimum sample size was calculated using the expression of medium effect formula

$$N = n \, (Z_1 - Z_2)^2 / ES^2$$

Where, N = minimum sample size; n = number of groups = 2; $Z_1$ = α-confidence interval at 0.05 = 1.96; $Z_2$ = β-confidence interval at 0.01 = 0.84; ES = medium effect = 0.21

Therefore, a minimum sample size of 356 participants consisting of 173 each of commercial bus drivers and conductors was projected to participate in this study.

### Instrument

**Nordic Musculoskeletal Questionnaire [13].**   This was used to assess Work-related Musculoskeletal Disorders (WMSDs). This questionnaire consists of structured, forced, binary scale that is self- administered. It has two sections: the first section contains identifying and anthropometric variables while the second section contains the musculoskeletal discomfort form, focused of specific body parts. The test–retest reliability of this instrument is ranges between 77–100% and the validity ranges between 80–100%.(13) It has a high specificity (0.71–0.88) and a high sensitivity (0.66–0.92) [14] This instrument has been used among healthy adult participants in this environment [5, 15]

**Driving Work Station Assessment (DWSA) form.**   This was used to retrieve relevant work related details. It was adapted after a brief modification of the workstation assessment proforma used by Ekechukwu et al [15], to suit the driving work environment. The form was adapted following a two-stage process. The first stage involved generation of new items and/or retrieval of unimportant items until saturation was achieved by eight experts. The second stage involved content validation by another group of eight experts, after which a content validity index was determined (CVI = 4.6). The DWSA form has three sections (A, B, and C). Section A assessed participants' demographic details, this section had 10 items. The second section assessed their job related variables and had 16 items while section C had 10 items and was used to assess the ergonomic details in their workstations. Thus the form had a total of 36items and a binary scaling.

**Stethoscope (Classic IITM Littman, USA) and Aneroid Sphygmomanometer (Homelife, Germany).**   These were used to assess the participants' systolic and diastolic blood pressures in millimeter mercury (mmHg).

**Stadiometer (HX-203 Portable Stadiometer, China).**   It was used to measure the heights of the participants in meters (m).

**Weighing scale (Harrison, China).**   It was used to measure the weight of the participants in kilogram (Kg).

**Tape rule (Shangha, China).**   It was used to measure the hip circumference, as well as waist circumference of the participants in meters (m).

## Ethics statement

Ethical approval was sought and obtained from the Health Research and Ethics Committee of the University of Nigeria Teaching Hospital (UNTH), Ituku-Ozalla, Enugu, Nigeria. The aim, purpose and relevance of the study were explained to the participants after being screened for eligibility. The informed consent form was given to those that voluntarily agreed to participate in the study. The individual in this manuscript has given written informed consent (as outlined in PLOS consent form) to publish these case details including the photograph.

## Procedure

The Demographic details of the eligible participants were first recorded while anthropometric variables such as height, weight, BMI, waist circumference, hip circumference, waist-hip ratio, conicity and abdominal volume indices, and cardiovascular variables such as systolic and diastolic blood pressures, pulse rate were assessed using standard protocols [16, 17]. The outcome measures (Nordic Musculoskeletal Questionnaire and Driving Workstation Assessment Form) were then administered.

The Nordic Musculoskeletal questionnaire (NMQ) and the Driving Work Station Assessment (DWSA) Form were randomly administered to the participant to prevent data setting. Each participant was made to sit comfortably, and was given the NMQ and MWSA to assess WMSD and work related details. These instruments are client-administered questionnaires, however, participants requiring further explanation on how to fill the questionnaire got the support of the research assistants. The DWSA elicited work related information from the participants such as their employment status (whether there were the mini bus owner or were paid employees), job satisfaction, job stress, work duration (average hours they work in a day), work frequency (average number of days they work in a week) as well as other WMSD risks such as constrained and akward postures, work pace and fatigue.

Some ergonomic concepts such as reach, clearance, awkward and constrained postures, symptoms of musculoskeletal disorders, job stress indicators etc were explained to the participants using lay terms. i. Reach was described as placement of object in a region that will not warrant unnecessary strectches inorder to use it; ii. Clearance was described as adequate workspace with no barriers on the way. Concept of adequate knee clearance was demonstrate; iii. Awkward posture was described as a non-neural posture that exerts more pressure on the body and examples were given using pictures such as described in Fig 2; iv. Constrained Postures was decribed as static or restricted posture that can impede blood flow; v. Symptoms of musculoskeletal disorders: early, intermediate and late symptoms of WMSD were described respectively as pains that disappear with a little rest while at work, pain that persists until rest at the close of work, and pain that persist after work closure and may cause sleep and leisure disturbances; vi. Job stress indicators such as high blood pressure, pulse rate, increased blinking, loss of concentration, fatigue etc were explained.

## Data analysis

Data obtained were cleaned and analyzed using SPSS version 21.0. Descriptive statistics of frequency, percentages, mean, and standard deviation were used to summarize the demographic, anthropometric and work related variables. Chi-square test used to assess the association between WMSD and selected variables. Binomial logistic regression model was used to predict the likelihood of the occurrence of WMSD. Level of significance was set at $\alpha = 0.05$.

**Table 1. Mean distribution of participants anthropometric and cardiovascular variables (N = 379).**

| Variable | Total Participants (n = 379) | | | | Mini Bus Drivers (n = 200) | | | | Mini Bus Conductors (n = 179) | | | |
|---|---|---|---|---|---|---|---|---|---|---|---|---|
| | Min | Max | Mean | SD | Min | Max | Mean | SD | Min | Max | Mean | SD |
| Age (years) | 20.00 | 66.00 | 33.26 | 10.76 | 21.00 | 66.00 | 38.42 | 10.22 | 20.00 | 60.00 | 27.50 | 8.13 |
| Height (m) | 1.49 | 1.87 | 1.72 | 0.05 | 1.58 | 1.87 | 1.72 | 0.05 | 1.49 | 1.86 | 1.72 | 0.05 |
| Weight (Kg) | 56.00 | 125.00 | 70.38 | 9.04 | 57.00 | 125.00 | 74.01 | 9.20 | 56.00 | 85.00 | 66.31 | 6.90 |
| WC (cm) | 62.00 | 124.00 | 88.26 | 11.97 | 62.00 | 124.00 | 92.30 | 11.26 | 62.00 | 118.00 | 83.73 | 11.12 |
| HC (cm) | 53.00 | 162.00 | 97.35 | 10.85 | 53.00 | 122.00 | 100.35 | 9.90 | 72.00 | 162.00 | 94.00 | 10.92 |
| BMI (Kg/m$^2$) | 10.66 | 40.35 | 23.83 | 3.21 | 10.66 | 40.35 | 24.97 | 3.38 | 19.35 | 35.58 | 22.56 | 2.45 |
| CI (kg$^2$m$^{-1}$) | 0.09 | 1.75 | 0.26 | 0.16 | 0.09 | 1.75 | 1.29 | 0.17 | 0.95 | 1.70 | 1.24 | 0.14 |
| AVI (cm$^3$) | 7.79 | 30.82 | 15.94 | 4.35 | 7.79 | 30.82 | 17.32 | 4.26 | 7.83 | 29.20 | 14.40 | 3.91 |
| SBP (mmHg) | 100.00 | 180.00 | 122.26 | 8.00 | 100.00 | 142.00 | 122.02 | 8.49 | 105.00 | 180.00 | 122.53 | 7.44 |
| DBP (mmHg) | 60.00 | 90.00 | 76.77 | 7.08 | 60.00 | 90.00 | 77.13 | 6.99 | 60.00 | 90.00 | 76.37 | 7.19 |
| PR (bpm) | 56.00 | 100.00 | 71.54 | 5.22 | 56.00 | 100.00 | 71.97 | 6.19 | 62.00 | 86.00 | 71.07 | 3.80 |

## Results

### Mean distribution of participants' anthropometric and cardiovascular variables

A total of 379 participants (200 commercial mini-bus drivers, and 179 mini-bus conductors) took part in this study. Their ages ranged from 20–66 years with a mean age of 33.26±10.76years (drivers = 38.42±10.22years, conductors = 27.50±8.13years). The mean BMI of the participants was 23.83±3.21kg/m$^2$. Also, the mean conicity index and abdominal volume index of the participants were 1.26±0.16 and 15.93±4.34 respectively. The participants in this study had a mean systolic and diastolic blood pressure of 122.2±8.0 mmHg and 76.77±7.08mmHg respectively. Their mean pulse rate was 71.54±5.21beats/min as shown in Table 1.

### Comparison of demographic and work related variables of commercial mini bus drivers and mini bus conductors

All the participants in this study were male, most of who were married (56.5%), had up to the Senior Secondary School level education (43.5%), did not smoke (54.1%), but took alcohol (84.4%). The comparison revealed that a significantly greater proportion of the mini bus drivers than the mini bus conductors were married ($X^2$ = 79.89, p<0.001), and owned their mini buses ($X^2$ = 60.89, p<0.001). Also, a significantly greater proportion ($X^2$ = 37.38, p<0.001) of mini-bus drivers had a tertiary education than the conductors. A good number of the participants reported experiencing severe job stress (73.4%) and also that they would like to change their job if given the opportunity (88.9%). Comparatively, a significantly higher proportion ($X^2$ = 8.53, p = 0.036) of the mini bus conductor experienced severe job stress than the mini bus drivers. Most of the participants worked 9-12hrs/day and for 6 or more days/week. A significantly greater proportion of the mini bus conductors than the drivers worked more than 8hours/day ($X^2$ = 11.49, p = 0.009) as shown in Table 2.

### Comparison of work related musculoskeletal disorders between commercial mini drivers and mini bus conductors

The general prevalence of work related musculoskeletal disorder (WMSD) among the participants was 95.8%. This general WMSD prevalence was higher among the mini bus conductors (97.8%) than the mini busdrivers (94.0%), though non-significantly ($X^2$ = 3.31, p = 0.069).

**Table 2. Frequency distribution of participants demographic and work related variables (N = 379).**

| Variables | Categories | Total Participants (n = 375) | | Mini Bus Drivers (n = 200) | | Mini Bus Conductors (n = 175) | | $X^2$ | p |
|---|---|---|---|---|---|---|---|---|---|
| | | f | % | f | % | f | % | | |
| Marital Status | Single | 165 | 44 | 44 | 22.0 | 121 | 67.6 | 79.894 | <0.001* |
| | Married | 214 | 57 | 156 | 78.0 | 58 | 32.4 | | |
| Smokers | ————————— | 174 | 46 | 99 | 49.5 | 75 | 41.9 | 2.197 | 0.138 |
| Alcohol Consumers | ————————— | 320 | 84 | 172 | 86.0 | 148 | 82.7 | 0.791 | 0.374 |
| Educational Status | None | 26 | 6.9 | 14 | 7.0 | 12 | 6.7 | 37.383 | <0.001* |
| | Primary school | 84 | 22 | 55 | 27.5 | 29 | 16.2 | | |
| | Junior Secondary | 60 | 16 | 20 | 10.0 | 40 | 22.3 | | |
| | Senior Secondary | 165 | 44 | 74 | 37.0 | 91 | 50.8 | | |
| | Tertiary | 44 | 12 | 37 | 18.5 | 7.0 | 4.0 | | |
| Employment status | Bus owner | 95 | 25 | 83 | 41.5 | 12 | 6.7 | 60.889 | <0.001* |
| | Employee | 284 | 75 | 117 | 58.5 | 167 | 93.3 | | |
| Sleep duration | < 4 hours | 1 | 0.3 | 1 | 0.5 | 0 | 0.0 | 0.900 | 0.638 |
| | 4–6 hours | 44 | 12 | 23 | 11.5 | 21 | 11.7 | | |
| | >6 hours | 334 | 88 | 176 | 88.0 | 158 | 88.3 | | |
| Job satisfaction | Satisfied | 259 | 68 | 144 | 72.0 | 115 | 64.2 | 2.625 | 0.105 |
| | Dissatisfied | 120 | 32 | 56 | 28.0 | 64 | 35.8 | | |
| Job stress | None | 5 | 1.3 | 4 | 2.0 | 1 | 0.6 | 8.531 | 0.036* |
| | Mild | 23 | 6.1 | 13 | 6.5 | 10 | 5.6 | | |
| | Moderate | 73 | 19 | 48 | 24.0 | 25 | 14.0 | | |
| | Severe | 278 | 73 | 135 | 67.5 | 143 | 79.9 | | |
| Change of Job | Yes | 337 | 89 | 173 | 86.5 | 164 | 91.6 | 2.513 | 0.113 |
| | No | 42 | 11 | 27 | 13.5 | 15 | 8.4 | | |
| Work duration | < 5 hours | 7 | 1.8 | 7 | 3.5 | 0 | 0.0 | 11.486 | 0.009* |
| (Hours/day) | 5–8 hours | 26 | 6.9 | 19 | 9.5 | 7 | 3.9 | | |
| | 9–12 hours | 219 | 58 | 109 | 54.5 | 110 | 61.5 | | |
| | > 12 hours | 127 | 34 | 65 | 32.5 | 62 | 34.6 | | |
| Work duration | < 5 days | 3 | 0.8 | 2 | 1.0 | 1 | 0.6 | 5.187 | 0.159 |
| (days/week) | 5 days | 7 | 1.8 | 5 | 2.5 | 2 | 1.1 | | |
| | 6 days | 329 | 87 | 178 | 89.0 | 151 | 84.4 | | |
| | 7 days | 41 | 11 | 15 | 7.5 | 25 | 14.0 | | |

However, mini-bus conductors (BC) more than the mini-bus drivers (BD) had significantly higher WMSD prevalence for the Neck ($X^2$ = 44.70, p<0.001) and upper back ($X^2$ = 94.89, p<0.001) regions. Conversely, the mini-bus drivers had significantly higher prevalence of elbow ($X^2$ = 7.46, p = 0.006), wrist ($X^2$ = 7.29, p = 0.007), lower back ($X^2$ = 63.52, p<0.001), and knee ($X^2$ = 15.60, p<0.001) WMSDs than the mini-bus conductors. A significantly higher proportion ($X^2$ = 128.37, p<0.001) of mini-bus conductors than drivers adopted awkward postures (BC = 39.7%, BD = 2.0%); however, more mini-bus driver adopted constrained postures than the mini-bus conductors (BD = 39.0%, BC = 2.8%) as shown in Table 3.

## Association between Work-Related Musculoskeletal Disorder (WMSD) and selected participants variables

There was a significant association between WMSD and each of work duration in hours/day ($X^2$ = 11.634, p = 0.009), and work frequency in days/week ($X^2$ = 8.394, p = 0.039). Also, there

**Table 3. Frequency distribution of WMSD characteristics of the participants (N = 379).**

| Variables | Categories | Total Participants (n = 379) | | Mini Bus Drivers (n = 200) | | Mini Bus Conductors (n = 179) | | $X^2$ | p |
|---|---|---|---|---|---|---|---|---|---|
| | | f | % | f | % | f | % | | |
| **General WMSD** | | 363 | 96 | 188 | 94.0 | 175 | 97.8 | 3.312 | 0.069 |
| **Trouble** | | 110 | 29 | 65 | 32.5 | 45 | 25.1 | 2.484 | 0.115 |
| **Regional WMSD** | Neck | 141 | 37 | 43 | 21.5 | 80 | 44.7 | 44.696 | <0.001* |
| | Shoulder | 179 | 47 | 99 | 49.5 | 80 | 44.7 | 0.876 | 0.349 |
| | Elbow | 30 | 7.9 | 23 | 11.5 | 7 | 3.9 | 7.464 | 0.006* |
| | Wrist. Hand | 27 | 7.1 | 21 | 10.5 | 6 | 3.4 | 7.294 | 0.007* |
| | Upper back | 205 | 54 | 61 | 30.5 | 144 | 80.4 | 94.894 | <0.001* |
| | Lower back | 253 | 67 | 170 | 85.0 | 83 | 46.4 | 63.52 | <0.001* |
| | Hip. Thighs | 18 | 4.7 | 13 | 6.5 | 5 | 2.8 | 2.869 | 0.090 |
| | Knees | 67 | 18 | 50 | 25.0 | 17 | 9.5 | 15.600 | <0.001* |
| | Ankle. Foot | 98 | 26 | 46 | 23.0 | 52 | 29.1 | 1.804 | 0.179 |
| **WMSD sites** | 1 | 75 | 20 | 38 | 19.0 | 22 | 12.3 | 4.974 | 0.083 |
| | 2–4 | 268 | 71 | 130 | 65.0 | 138 | 77.1 | | |
| | > 4 | 36 | 9.6 | 21 | 10.5 | 15 | 8.4 | | |
| **Risk Factors** | Constrained Posture | 83 | 22 | 78 | 39.0 | 5 | 2.8 | 128.371 | <0.001* |
| | Awkward Posture | 75 | 20 | 4 | 2.0 | 71 | 39.7 | | |
| | High Work Pace | 75 | 20 | 35 | 17.5 | 40 | 22.3 | | |
| | Fatigue | 90 | 24 | 47 | 23.5 | 43 | 24.0 | | |

was a significant association between WMSD and each of job satisfaction ($X^2$ = 10.620, P = 0.001), and job stress ($X^2$ = 16.879, p = 0.001). In the same vein, there was a significant association between WMSD and perceived cause of WMSD ($X^2$ = 70.428, P < 0.0001) as shown in Table 4.

## Prediction model for work related musculoskeletal disorders

A logistic regression was performed to ascertain the effects of age, pulse rate, daily work duration, weekly work frequency, job satisfaction, job stress, and perceived cause of the MSD on the likelihood that the participants have WMSD within a 12month period. The logistic regression model was statistically significant ($X^2$ = 149.35; p <0.0001). The model explained 66.0% of the variance in WMSD and correctly classified 97.1% of cases. The model revealed that working for greater than 4days in a week was a significant predictor of WMSD in this population. Also, job dissatisfaction significantly increases the odds of WMSD occurrence (OR = 1.990, p = 0.031). The model also showed that certain factors such as constrained posture (OR = 5.324, p = 0.003), and fatigue (OR = 4.719, p = 0.002) had significant effects on the occurrence of WMSD among commercial mini bus drivers and driver assistants as shown in Table 5.

## Discussion

Globally, work associated musculoskeletal disorder is a menace that not only affects the economy in terms of reduced productivity; but also adversely distorts health and quality of life of the affected persons. About nine of every ten participants in this study had a WMSD in at least one body region. A Similarly high WMSD prevalence (97.9%) was reported by Aini and Huda [18], in a study among drivers in the University of Malaysia. Contrarily, studies by Sekkay et al

**Table 4. Association between MSD and selected participants variables (N = 379).**

| Variables | Categories | f(n) | % | X² | P |
|---|---|---|---|---|---|
| Marital status | Single | 159(165) | 96.36 | 0.248 | 0.619 |
| | Married | 204(214) | 95.32 | | |
| Educational status | None | 26(26) | 100 | 4.669 | 0.700 |
| | Primary | 79(84) | 94.05 | | |
| | JSS | 59(60) | 98.33 | | |
| | SSS | 158(165) | 95.75 | | |
| | NCE | 15(16) | 93.75 | | |
| | OND | 19(21) | 90.47 | | |
| | HND | 5(5) | 100 | | |
| | BSC | 2(2) | 100 | | |
| Alcohol consumption | Yes | 307(320) | 95.93 | 0.129 | 0.720 |
| | No | 56(59) | 94.91 | | |
| Smoking | No | 198(205) | 96.58 | 0.719 | 0.396 |
| | Yes | 165(174) | 94.82 | | |
| Sleep | <4 hours | 1(1) | 100 | 0.056 | 0.972 |
| | 4–6 hours | 42(44) | 95.45 | | |
| | >6 hours | 320(334) | 95.80 | | |
| Work duration (hours/day) | 1-4hours | 5(7) | 71.42 | 11.634 | 0.009* |
| | 5–8 hours | 24(26) | 92.30 | | |
| | 9–12 hours | 212(219) | 96.80 | | |
| | >12 hours | 122(127) | 96.06 | | |
| Work frequency (days/week) | 4 | 2(3) | 66.66 | 8.394 | 0.039* |
| | 5 | 6(7) | 85.71 | | |
| | 6 | 316(329) | 96.04 | | |
| | 7 | 39(40) | 97.5 | | |
| Job satisfaction | Satisfied | 254(259) | 98.06 | 10.620 | 0.001* |
| | Dissatisfied | 109(120) | 90.83 | | |
| New job | Yes | 323(337) | 95.84 | 0.034 | 0.853 |
| | No | 40(42) | 95.23 | | |
| Job stress | None | 5(5) | 100 | 16.879 | 0.001* |
| | Mild | 20(23) | 86.95 | | |
| | Moderate | 65(73) | 89.04 | | |
| | Severe | 273(278) | 98.20 | | |
| Employment status | Owner | 91(95) | 95.78 | 0.082 | 0.995 |
| | Employee | 272(284) | 95.77 | | |
| Risk Factors | Constrained posture | 82(83) | 98.79 | 70.428 | <0.0001* |
| | Awkward posture | 75(75) | 100 | | |
| | High pace of work | 75(75) | 100 | | |
| | Fatigue | 89(90) | 98.88 | | |

*Key*: f (n) = number of participants with WMSDs (total number of participants); % = f/n(100)

* = significant.

[19], and Kärmeniemi et al [20], reported a lower WMSD prevalence among commercial drivers in Canada and Finland respectively. Differences in study design, economic and cultural ddifferences may have accounted for these disparities. This study also found that WMSD prevalence was higher among the mini bus conductors than the mini bus drivers, though non-

Table 5. Prediction model for musculoskeletal disorders of the participants (N = 379).

| Predictors | Categories | Model summary | | OR | P |
|---|---|---|---|---|---|
| | | X² (ρ) | R² (C) | | |
| Age | | . . . . . . . . . . . 149.35 (<0.0001) . . . . . . . . . | . . . . . . . . . . 0.660 (97.1%) . . . . . . . . . . | 0.066 | 0.063 |
| Pulse Rate | | | | 0.099 | 0.150 |
| Daily work duration | 5-8hrs | | | -0.718 | 0.738 |
| | 9-12hrs | | | -0.985 | 0.619 |
| | >12hrs | | | -0.518 | 0.790 |
| Work day per week | 5 days | | | 10.019 | 0.001* |
| | 6 days | | | 9.176 | <0.001* |
| | 7 days | | | 9.240 | 0.010* |
| Job dissatisfaction | | | | 1.990 | 0.031* |
| Job stress | Mild | | | 23.475 | 0.999 |
| | Moderate | | | 22.588 | 0.999 |
| | Severe | | | 20.062 | 0.999 |
| Perceived causes | Constrained posture | | | 5.324 | 0.003* |
| | Akward posture | | | 19.442 | 0.996 |
| | High work pace | | | 19.086 | 0.996 |
| | Fatique | | | 4.719 | 0.002* |

Key: X² = Chi-square value; ρ = Level of significance of the equation (Model)

R² = Nagelkerke R-square; C = Overall classification; OR = Odds ratio

* = Significant.

significantly. Although there appear to be no single study that compared WMSD prevalence between mini bus conductors and drivers, the reported WMSD prevalence among Indian bus conductors (93.3%) by Gangopadhyay et al [21], is higher than those of Indian bus drivers (85.0%) as reported by Borle et al [22]. The work tasks of bus conductors require that they stand often, sometimes in awkward postures coupled with the high work intensities they are often exposed to [21]. These factors may have therefore been responsible for the greater prevalence of WMSD among bus conductors than their driver counterparts in this study. There is an urgent need for ergonomic training and intervention for both bus drivers and conductors tailored to address this public health challenge. However, this may be done alongside a review of the occupational policies and regulations in this industry as well as their proper implementation.

Most of the commercial mini bus drivers and mini bus conductors in this study worked (on-duty time) for more than 9 hours per day and a good number worked even beyond 12 hours per day at a work frequency not less than 6 days per week. Based on the 1984 Occupational Safety and Health (OSH) Act 3.132 [23], A commercial vehicle driver (CVD) must take at least 20minutes of break during every 5hours of work time (including at least 10 consecutive minutes during or at the end of 5hours). In addition, he/she is expected to observe not more than 168 hours of work time in any 14day period [23]. This implies that for the participants in this study that worked beyond 12hours per day (33.5%) for 7days/week (10.7%), their on-duty time in consecutive 14days is beyond 182 hours (greater than 168 maximum recommended by OSH). The lower back and upper back (spine) were the body regions most commonly affected by WMSD. This finding is similar to the reports of studies conducted in Taiwan and Malaysia [24, 25]. They both found high prevalence of low back pain among drivers.

There was a significant association between WMSD and each of duration of working (hours per day) and work frequency (days per week). However, only work frequency was a

significant predictor of WMSD. This implies that while working for longer hours in a day and/ or more days in a week may be associated with the risk of WMSD among drivers and their assistants, working for more than 4days in a week increases the likelihood of developing WMSDs by over 9folds in this population. The observed association between working hours and WMSDs conforms with previous studies among drivers in Malaysia and South Africa [18, 25, 26]. Biomechanical task analysis of driving would reveal that it is a task characterized by prolonged sitting posture in which the body weight is transmitted to the pelvis (ischial tuberosities) through the lumbar spine [27]. Also, the axis for upper bodily movements in sitting runs through the lumbar vertebrae which incidentally are the least stable regions of the human spine. The risk of developing WMSD in this region is further heightened by the poor ergonomic designs of the seats found in most commercial mini buses that hardly support this very mobile, yet unstable region of the body. Therefore proper ergonomic education and early evaluation of commercial mini bus drivers are recommended to address this problem.

Prolonged constrained and awkward posture inside the mini bus may be responsible for the WMSDs among mini bus conductors. In a bid to maximise financial returns, most mini bus conductors in Nigeria have a common practice of not reserving a seat for themselves in the mini bus. They rather would accommodate more passengers in their seat while they stand in the mini bus with their trunk flexed and tucked forward (a typical posture adopted by mini bus conductors). They often assume this constrained, awkward posture for most part of their work duration in a day and for the many years they may have been on the job. Also in a bid to make more profit, drivers and their assistants work for 6days in a week and sometimes all the days of the week as reported in this study. Working at this rate strains the muscles giving no adequate rest time for proper tissue repair and recovery. This cumulative trauma results in the development of WMSDs in this population. However, this is contrary to the report by Akinpelu et al [2], that there is no significant association between WMSD and work duration among bus drivers in Ibadan. This difference in report may be due to difference in research design and location. While the study by Akinpelu et al [2], recruited only bus drivers, the present study had both bus drivers and their assistants. Also, the work practice and job strain in Ibadan may not be the same as obtainable in Enugu. However more studies are recommended in other location to further elucidate on these findings in this field of study.

In this study, a significant association between job stress and WMSD was found; and further analysis showed that perceived job stress was a significant predictor of WMSD. This association between WMSDs among commercial mini bus drivers and mini bus conductors with a high level of self-rated job stress is consistent with various studies [4, 28–30]. The study by Kim et al [28], revealed a significant association between WMSD and job stress. There was also a significant relationship found between job stress dimensions and WMSD among other job areas like Physiotherapy [29]. Job stress has been shown to be the cumulative effects of the factors such as high work pace/pressure [31], constrained and awkward posture [15], and work strain [32]; which were reported to be found among the participants of this study.

There was no significant association between WMSD and variables such as marital status, educational status, alcohol and smoking habit. This finding is consistent with the report by Aini and Huda [18], that equally found no significant association between WMSD and each of marital status, educational level, BMI, alcohol and smoking habit, duration of employment and sport activities. There was also no statistically significant association between age and WMSD. Similar findings were reported in previous studies [33–35]. In contrast, Naidoo et al [36], showed that older workers are more likely to report WMSD than younger workers. Also Abolfazl et al [37], further highlighted the importance of age in the development of musculoskeletal disorders. A major limitation of this study is selection bias due to the non-probability

sampling technique use. However attempts were made to control for this using regression modeling.

## Conclusion

Based on the outcome of this study, we conclusively state that there is a high prevalence of WMSDs among commercial mini-bus drivers and driver assistants (conductors). Also, there is a significant association between WMSD and each of work duration, work frequency, job satisfaction, job stress and perceived cause of MSD. Working for more than 4days in a week, job dissatisfaction, job stress, constrained posture and fatigue are significant predictors of WMSD among commercial mini-bus drivers and driver assistants (conductors).

## Author Contributions

**Conceptualization:** Echezona Nelson Dominic Ekechukwu, Erobogha Useh, Victor Adimabua Utti.

**Data curation:** Erobogha Useh, Ukachukwu Okaroafor Abaraogu.

**Formal analysis:** Obumneme Linky Nna, Sussan Uzoamaka Arinze-Onyia.

**Investigation:** Ogbonna Nnajiobi Obi, Sussan Uzoamaka Arinze-Onyia.

**Methodology:** Echezona Nelson Dominic Ekechukwu, Nmachukwu Ifeoma Ekechukwu, Emmanuel Nwabueze Aguwa.

**Project administration:** Erobogha Useh.

**Resources:** Obumneme Linky Nna, Nmachukwu Ifeoma Ekechukwu.

**Supervision:** Echezona Nelson Dominic Ekechukwu, Emmanuel Nwabueze Aguwa.

**Validation:** Ogbonna Nnajiobi Obi.

**Writing – original draft:** Echezona Nelson Dominic Ekechukwu, Obumneme Linky Nna, Nmachukwu Ifeoma Ekechukwu.

**Writing – review & editing:** Echezona Nelson Dominic Ekechukwu, Ukachukwu Okaroafor Abaraogu, Victor Adimabua Utti.

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
