## [Decision Letter · Decision Letter 0]

7 May 2021

PONE-D-21-05678

Ergonomic Assessment of Work-Related Musculoskeletal Disorder and its Determinants among Commercial Bus Drivers and Driver Assistants (Bus Conductors) in Nigeria

PLOS ONE

Dear Dr. EKECHUKWU,

Thank you for submitting your manuscript to PLOS ONE. After careful consideration, we feel that it has merit but does not fully meet PLOS ONE’s publication criteria as it currently stands. Therefore, we invite you to submit a revised version of the manuscript that addresses the points raised during the review process.

We look forward to receiving your revised manuscript.

Kind regards,

Matias Noll, Ph.D

Academic Editor

PLOS ONE

Journal Requirements:

2)  We note that Figures 1 to 3 includes an image of a participant in the study.

Reviewers' comments:

Reviewer's Responses to Questions

**Comments to the Author**

1. Is the manuscript technically sound, and do the data support the conclusions?

Reviewer #1: Yes

Reviewer #2: Yes

Reviewer #3: Yes

2. Has the statistical analysis been performed appropriately and rigorously? 

Reviewer #1: No

Reviewer #2: Yes

Reviewer #3: Yes

3. Have the authors made all data underlying the findings in their manuscript fully available?

Reviewer #1: No

Reviewer #2: Yes

Reviewer #3: No

4. Is the manuscript presented in an intelligible fashion and written in standard English?

Reviewer #1: Yes

Reviewer #2: Yes

Reviewer #3: Yes

5. Review Comments to the Author

Reviewer #1: This article evaluates work-related musculoskeletal disorder and its determinants among Nigerian commercial bus driver and bus conductors. The topic is important, however, the presentation and writing of the article needs to be substantially improved. Also, some key information and results should be explained in detail.

1. The article lacks a clearly formulated research question and hypotheses. The choice of measures and the report of the results is therefore difficult to follow and seems arbitrary.

2. The literature review is sparse. Could the authors include more literature on biomechanic assessment that could be used to give precise recommendations as to how the job or the spatial constraints inside a bus would need to change to improve the working situation of the drivers and conductors?

3. More information about the procedure would be appreciated. How exactly where the interviews conducted?

4. The authors claim that there is an urgent need for ergonomic training and intervention for both bus drivers and conductors tailored to address this public health challenge. Is this a feasible countermeasure or does the overall occupational, legal situation need to change? Could the authors derive more specific guidelines on what needs to change?

5. Formatting: The article lacks the necessary line numbers. This makes it difficult to provide detailed annotations. The citation style does not meet PLOS requirements and is partially misplaced.

6. The relevant data are not within the manuscript and its supporting Information files.

7. The authors should elaborate more on the rationale for their measures. E.g., why was a pulse rate measured and reported? Pulse rate is a very instantaneous measure.

8. Prediction Model: Is it reasonable to predict the probability of receiving MSDs for participants who already show such symptoms. Could the authors elaborate on why they chose this statistical method? Should the dataset be divided into participants who already show MSD and those who do not?

9. Figure 1 is blurred and not properly selected to represent the authors’ intent (e.g., the back support is barely visible). Why is it important that the driver's neck is flexed in this figure? How is this consistent with the figure description in the text?

Reviewer #2: This paper presents study that assess the determinants of musculoskeletal disorders among commercial bus drivers and bus conductors in Nigeria with consideration of an ergonomics issues. The paper presents interesting results, however presentation of the study needs corrections.

Introduction:

Second paragraph describes bus conductors work. It does not belong to Introduction section. In this section would be expected justification of the study with presentation of hypothesis and research questions, which is missing.

There is lack of presentation of the research problem, that the presented study want to solve.

Method

2.1. participants: “Thus, N = 2(1.96 +0.84)2 / (0.21)2 = 15.68 / 0.0441 = 355.56” – presenting this calculations is too trivial.

“Some ergonomic concepts such as reach, clearance, awkward and constrained postures, symptoms of musculoskeletal disorders, job stress indicators etc were explained to the participants using lay terms.” – how those concepts were defined?

Results

Results are presented in tables. The same time a lot of those data values are repeated in the text. It is not excessive and unnecessary?

“Prediction Model for Work Related Musculoskeletal Disorders” – why the equation (model) is not presented?

Discussion of study limitations is missing.

Reviewer #3: 1. Indication of minibus drivers as bus drivers is not a proper comparison, these concepts should be detailed, perhaps also in the topic of the article, the differences between the work of a bus driver and a minibus are indirectly confirmed by the research cited in the article: "Contrarily, studies by Sekkay et al, (16) ) and Kärmeniemi et al, (17) reported a lower WMSD prevalence among commercial drivers in Canada and Finland respectively. "

2. Average parameters of the heart work indicate a healthy group of people in this respect, also taking into account the age of the respondents, which does not confirm musculoskeletal problems, especially in over 95% of employees.

3. It is worth emphasizing in the article that most of the respondents worked 6 or more days a week, only 10 of the respondents worked 4 or 5 days a week.

6. PLOS authors have the option to publish the peer review history of their article (what does this mean?). If published, this will include your full peer review and any attached files.

Reviewer #1: No

Reviewer #2: No

Reviewer #3: No

---

## [Author Response · Author response to Decision Letter 0]

5 Jul 2021

Editor, PLOS One

Dear Editor:

Thank you for your interest in our manuscript (PONE-D-21-05678) titled, "Ergonomic Assessment of Work-Related Musculoskeletal Disorder and its Determinants among Commercial Mini Bus Drivers and Driver Assistants (Mini Bus Conductors) in Nigeria". We have carefully read your comments and recommendations and find them to be helpful. We have taken great care to present our commentary with accuracy, precision, and clarity.

To facilitate your evaluation of our revised manuscript, we have incorporated all the reviewers’ comments (italicized) in our reply (bold) and addressed the requests with specific point-by-point responses to each query. We separately delineated and numbered each query with a corresponding response. All editions, additions and changes in the manuscript using the MS Word Track change Highlight. We also indicated the precise location in the revised manuscript where we have addressed your comments. All references to the location of our changes (i.e., page) correspond to the revised manuscript.

Reviewer #1: 

This article evaluates work-related musculoskeletal disorder and its determinants among Nigerian commercial bus driver and bus conductors. The topic is important, however, the presentation and writing of the article needs to be substantially improved. Also, some key information and results should be explained in detail.

Response: Thank you for your review and finding the topic important. We have made effort to improve the presentation and writing of the article. Details of key information and results have been equally explained in detail. 

i. The article lacks a clearly formulated research question and hypotheses. The choice of measures and the report of the results is therefore difficult to follow and seems arbitrary.

Response: Attempts have been made to clearly formulate the research question and hypotheses that will hopefully make the choice of measure and report of the result easier to follow and understand (See page 3)

ii. The literature review is sparse. Could the authors include more literature on biomechanic assessment that could be used to give precise recommendations as to how the job or the spatial constraints inside a bus would need to change to improve the working situation of the drivers and conductors?

Response: Further literature review on the biomechanics of prolonged sitting and standing and how they relate to MSD pathology has been done (See page 2)

iii. More information about the procedure would be appreciated. How exactly where the interviews conducted?

Response: This study had a quantitative design, there was no qualitative interviews. Also, the instruments used in this study were client-administered. Only minimal guidance in understanding how to fill these instruments were given. This information has also been provided (See page 6)

iv. The authors claim that there is an urgent need for ergonomic training and intervention for both bus drivers and conductors tailored to address this public health challenge. Is this a feasible countermeasure or does the overall occupational, legal situation need to change? Could the authors derive more specific guidelines on what needs to change?

Response: Thank you for this very important suggestion. Changing the overall occupational and legal situation is a possible approach but may take a much longer time to be implemented. More so, the targeted persons may not understand the merits behind it and may result in the problem of compliance. However, education should be the first line of action before or alongside this important suggestion. This has been added to the discussion (See page 14)

v. Formatting: The article lacks the necessary line numbers. This makes it difficult to provide detailed annotations. The citation style does not meet PLOS requirements and is partially misplaced.

Response: Sorry about the difficulty. This has been done.

vi. The relevant data are not within the manuscript and its supporting Information files.

Response: Pardon me, I am not sure I understood what you meant by the above statement. Kindly let me know the particular missing relevant data and supporting information 

vii. The authors should elaborate more on the rationale for their measures. E.g., why was a pulse rate measured and reported? Pulse rate is a very instantaneous measure.

Response: Some physiological variables such as pulse rate, blood pressure etc are health indices and so can be relied upon to make inference on the health status of the participants. These variables sometimes indicate the level of mental stress the person is exposed to. These are relevant vital signs for health.

viii. Prediction Model: Is it reasonable to predict the probability of receiving MSDs for participants who already show such symptoms. Could the authors elaborate on why they chose this statistical method? Should the dataset be divided into participants who already show MSD and those who do not?

Response: The prediction model was used to identify the important ergonomic determinants of WMSD in these cohorts. This result can be relied upon subsequently in preventing further occurrence or its worsening. The identified predictors can also be used in risk/hazard analysis and for education as a preventive measure.

Dividing the dataset into those with MSD and those without would imply a case-control design and will require a different analysis altogether even though this is not our design and objective. 

ix. Figure 1 is blurred and not properly selected to represent the authors’ intent (e.g., the back support is barely visible). Why is it important that the driver's neck is flexed in this figure? How is this consistent with the figure description in the text?

Response: Thanks for your observation. This was a still picture taken from a driver who doubled as a conductor. He was making efforts to communicate with some passengers and this forced him into some awkward posture. This practice is not uncommon in the study environment and so the picture attempts to further elucidate on the research problem (see page 2, line 13)

Reviewer #2: 

This paper presents study that assess the determinants of musculoskeletal disorders among commercial bus drivers and bus conductors in Nigeria with consideration of an ergonomics issues. The paper presents interesting results, however presentation of the study needs corrections.

Response: Thanks, we are very much willing to do so where necessary.

Introduction:

Second paragraph describes bus conductors work. It does not belong to Introduction section. In this section would be expected justification of the study with presentation of hypothesis and research questions, which is missing. There is lack of presentation of the research problem, that the presented study want to solve.

Response: This section gave the background of the study. It will be necessary to create this background for a better appreciation of the research problem and subsequent justification of this study. The research hypothesis/question has however been provided. Thank for the observation. (see page 3, lines 22-26). 

The research problem is contained in page 3 but more vividly stated in lines 18-22

Method

2.1. participants: “Thus, N = 2(1.96 +0.84)2 / (0.21)2 = 15.68 / 0.0441 = 355.56” – presenting this calculations is too trivial.

Response: This has been removed. Thanks for the observation.

“Some ergonomic concepts such as reach, clearance, awkward and constrained postures, symptoms of musculoskeletal disorders, job stress indicators etc were explained to the participants using lay terms.” – how those concepts were defined?

Response: These definitions have been added to the study (page 6)

Results

Results are presented in tables. The same time a lot of those data values are repeated in the text. It is not excessive and unnecessary?

“Prediction Model for Work Related Musculoskeletal Disorders” – why the equation (model) is not presented?

Response: The unnecessary repetitions have been deleted. Thanks for the observation

Logistic regression was done to determine the odds of developing WMSD and the model has been described and shown in table 5. This is unlike a multiple regression or a linear regression cannot be represented by a linear equation.

Discussion of study limitations is missing.

Response: Thanks, this has been added (see page 17)

Reviewer #3: 

1. Indication of minibus drivers as bus drivers is not a proper comparison, these concepts should be detailed, perhaps also in the topic of the article, the differences between the work of a bus driver and a minibus are indirectly confirmed by the research cited in the article: "Contrarily, studies by Sekkay et al, (16) ) and Kärmeniemi et al, (17) reported a lower WMSD prevalence among commercial drivers in Canada and Finland respectively. "

Response: Thanks, this change has been made

2. Average parameters of the heart work indicate a healthy group of people in this respect, also taking into account the age of the respondents, which does not confirm musculoskeletal problems, especially in over 95% of employees.

Response: Heart rate is not a sole determinant of health (a multifaceted construct). Also younger persons are able to accommodate both physical and mental workload without a significant change in heart rate. Finally, heart rate and age were not found to be predictive of WMSD. 

3. It is worth emphasizing in the article that most of the respondents worked 6 or more days a week, only 10 of the respondents worked 4 or 5 days a week.

Response: Thanks for this insight. It has been emphasized (page 7)

Overall, we are very pleased with our revised manuscript based on the recommendations of the reviewers. We thank you for your interest and continued consideration of our application

Most respectfully, 

Dr. Ekechukwu E.N.D

Department of Medical Rehabilitation, FHST,

College of Medicine, University of Nigeria.

(nelson.ekechukwu@unn.edu.ng)

---

## [Decision Letter · Decision Letter 1]

17 Aug 2021

PONE-D-21-05678R1

Ergonomic Assessment of Work-Related Musculoskeletal Disorder and its Determinants among Commercial Mini Bus Drivers and Driver Assistants (Mini Bus Conductors) in Nigeria

PLOS ONE

Dear Dr. EKECHUKWU,

Thank you for submitting your manuscript to PLOS ONE. After careful consideration, we feel that it has merit but does not fully meet PLOS ONE’s publication criteria as it currently stands. Therefore, we invite you to submit a revised version of the manuscript that addresses the points raised during the review process.

We look forward to receiving your revised manuscript.

Kind regards,

Matias Noll, Ph.D

Academic Editor

PLOS ONE

Journal Requirements:

Reviewers' comments:

Reviewer's Responses to Questions

**Comments to the Author**

1. If the authors have adequately addressed your comments raised in a previous round of review and you feel that this manuscript is now acceptable for publication, you may indicate that here to bypass the “Comments to the Author” section, enter your conflict of interest statement in the “Confidential to Editor” section, and submit your "Accept" recommendation.

Reviewer #1: (No Response)

Reviewer #2: All comments have been addressed

Reviewer #3: All comments have been addressed

2. Is the manuscript technically sound, and do the data support the conclusions?

Reviewer #1: Yes

Reviewer #2: Yes

Reviewer #3: Yes

3. Has the statistical analysis been performed appropriately and rigorously? 

Reviewer #1: Yes

Reviewer #2: Yes

Reviewer #3: Yes

4. Have the authors made all data underlying the findings in their manuscript fully available?

Reviewer #1: No

Reviewer #2: Yes

Reviewer #3: No

5. Is the manuscript presented in an intelligible fashion and written in standard English?

Reviewer #1: Yes

Reviewer #2: Yes

Reviewer #3: Yes

6. Review Comments to the Author

Reviewer #1: 1. I suggest that the hypotheses are presented in a list to make them more visible and to achieve a clear structure.

2. Why does the hypothesis expect a higher prevalence of WMSD among bus drivers rather than bus conductors? Please explain in more detail. This seems confusing because the literature review stated that bus conductors have even worse working postures. Shouldn’t they be the ones showing higher prevalence for WMSD?

3. The second hypothesis talks about "these cohorts". Which cohorts are meant here? Bus drivers, conductors of both? Please make this more clear.

Reviewer #2: (No Response)

Reviewer #3: I'm not sure if I understand the wording right "Commercial Mini Drivers". Please check if the change in the description of the studied group of people has been taken into account throughout the article, eg in conclusions.

7. PLOS authors have the option to publish the peer review history of their article (what does this mean?). If published, this will include your full peer review and any attached files.

Reviewer #1: No

Reviewer #2: No

Reviewer #3: No

---

## [Author Response · Author response to Decision Letter 1]

30 Sep 2021

Dear Editor:

Thank you for your interest in our manuscript (PONE-D-21-05678) titled, "Ergonomic Assessment of Work-Related Musculoskeletal Disorder and its Determinants among Commercial Mini Bus Drivers and Driver Assistants (Mini Bus Conductors) in Nigeria". We have carefully read your comments and recommendations and find them to be helpful. We have taken great care to present our commentary with accuracy, precision, and clarity.

To facilitate your evaluation of our revised manuscript, we have incorporated all the reviewers’ comments (italicized) in our reply (bold) and addressed the requests with specific point-by-point responses to each query. We separately delineated and numbered each query with a corresponding response. All editions, additions and changes in the manuscript were done using the MS Word Track change Highlight. We also indicated the precise location in the revised manuscript where we have addressed your comments. All references to the location of our changes (i.e., page) correspond to the revised manuscript.

1. If the authors have adequately addressed your comments raised in a previous round of review and you feel that this manuscript is now acceptable for publication, you may indicate that here to bypass the “Comments to the Author” section, enter your conflict of interest statement in the “Confidential to Editor” section, and submit your "Accept" recommendation.

Reviewer #1: (No Response)

Reviewer #2: All comments have been addressed

Reviewer #3: All comments have been addressed

Response: Thank you.

2. Is the manuscript technically sound, and do the data support the conclusions?

Reviewer #1: Yes

Reviewer #2: Yes

Reviewer #3: Yes

Response: Thank you.

3. Has the statistical analysis been performed appropriately and rigorously? 

Reviewer #1: Yes

Reviewer #2: Yes

Reviewer #3: Yes

Response: Thank you.

4. Have the authors made all data underlying the findings in their manuscript fully available?

Reviewer #1: No

Reviewer #2: Yes

Reviewer #3: No

Response: Thank you, a data repository link (https://data.mendeley.com/drafts/brt3myjxbm) as well as citation (EKECHUKWU, Echezona Nelson Dominic (2021), “WMSD among Bus Drivers and Conductors”, Mendeley Data, V1, doi: 10.17632/brt3myjxbm.1) for the data have been provided (See page 20, lines 9-12)

5. Is the manuscript presented in an intelligible fashion and written in standard English?

Reviewer #1: Yes

Reviewer #2: Yes

Reviewer #3: Yes

Response: Thank you.

6. Review Comments to the Author

Reviewer #1: 1. I suggest that the hypotheses are presented in a list to make them more visible and to achieve a clear structure.

2. Why does the hypothesis expect a higher prevalence of WMSD among bus drivers rather than bus conductors? Please explain in more detail. This seems confusing because the literature review stated that bus conductors have even worse working postures. Shouldn’t they be the ones showing higher prevalence for WMSD?

3. The second hypothesis talks about "these cohorts". Which cohorts are meant here? Bus drivers, conductors of both? Please make this more clear.

Response: Thank you.

1. The hypotheses have been presented in a list as suggested (see page 3, lines 21 and 22)

2. It is only a hypothesis, it can swing in any direction. However, this has been re-stated to make it less confusing. (see page 3, lines 22 and 23)

3. The cohort here refers to bus drivers and bus conductors. This has been restated for clarity (see page 3, line 23)

Reviewer #2: (No Response)

Response: None

Reviewer #3: I'm not sure if I understand the wording right "Commercial Mini Drivers". Please check if the change in the description of the studied group of people has been taken into account throughout the article, eg in conclusions.

Response: Thank you. The term “mini bus” was suggested inorder to further delineate the type of bus operated by the participants. Your comment on inconsistent use is well noted and appreciated. These have been corrected throughout the manuscript.

7. PLOS authors have the option to publish the peer review history of their article (what does this mean?). If published, this will include your full peer review and any attached files.

Do you want your identity to be public for this peer review? For information about this choice, including consent withdrawal, please see our Privacy Policy.

Reviewer #1: No

Reviewer #2: No

Reviewer #3: No

Response: None

Overall, we are very pleased with our revised manuscript based on the recommendations of the reviewers. We thank you for your interest and continued consideration of our application

---

## [Editor Report · Decision Letter 2]

5 Nov 2021

Ergonomic Assessment of Work-Related Musculoskeletal Disorder and its Determinants among Commercial Mini Bus Drivers and Driver Assistants (Mini Bus Conductors) in Nigeria

PONE-D-21-05678R2

Dear Dr. EKECHUKWU,

We’re pleased to inform you that your manuscript has been judged scientifically suitable for publication and will be formally accepted for publication once it meets all outstanding technical requirements.

Kind regards,

Matias Noll, Ph.D

Academic Editor

PLOS ONE

---

## [Editor Report · Acceptance letter]

19 Nov 2021

PONE-D-21-05678R2 

Ergonomic Assessment of Work-Related Musculoskeletal Disorder and Its Determinants among Commercial Mini Bus Drivers and Driver Assistants (Mini Bus Conductors) in Nigeria 

Dear Dr. Ekechukwu:

I'm pleased to inform you that your manuscript has been deemed suitable for publication in PLOS ONE. Congratulations! Your manuscript is now with our production department. 

Kind regards, 

on behalf of

Dr. Matias Noll 

Academic Editor

PLOS ONE